# Treatment and Systemic Sclerosis Interstitial Lung Disease Outcome: The Overweight Paradox

**DOI:** 10.3390/biomedicines10020434

**Published:** 2022-02-13

**Authors:** Alexandra Nagy, Erik Palmer, Lorinc Polivka, Noemi Eszes, Krisztina Vincze, Eniko Barczi, Aniko Bohacs, Adam Domonkos Tarnoki, David Laszlo Tarnoki, György Nagy, Emese Kiss, Pal Maurovich-Horvat, Veronika Müller

**Affiliations:** 1Department of Pulmonology, Semmelweis University, 1083 Budapest, Hungary; nagy.alexandra0718@gmail.com (A.N.); palmerperik@gmail.com (E.P.); polivka.lorinc@outlook.com (L.P.); noemi.eszes@gmail.com (N.E.); vinczekrisztina@gmail.com (K.V.); eniko.barczi@gmail.com (E.B.); bohacs.aniko@med.semmelweis-univ.hu (A.B.); 2Medical Imaging Centre, Semmelweis University, 1082 Budapest, Hungary; tarnoki2@gmail.com (A.D.T.); tarnoki4@gmail.com (D.L.T.); maurovich-horvat.pal@med.semmelweis-univ.hu (P.M.-H.); 3Department of Genetics, Cell- and Immunobiology, Semmelweis University, 1082 Budapest, Hungary; gyorgyngy@gmail.com; 4Department of Rheumatology and Clinical Immunology, Department of Internal Medicine and Oncology, Semmelweis University, 1083 Budapest, Hungary; 5Department of Clinical Immunology, Adult and Pediatric Rheumatology, National Institute of National Institute of Locomotor Diseases and Disabilities, 1023 Budapest, Hungary; drkiss.emese@gmail.com; 63rd Department of Internal Medicine and Haematology, Semmelweis University, 1088 Budapest, Hungary

**Keywords:** systemic sclerosis, interstitial lung disease, progressive fibrosing interstitial lung disease, lung function, body mass index, treatment

## Abstract

(1) Background: Systemic sclerosis (SSc) is frequently associated with interstitial lung diseases (ILDs). The progressive form of SSc-ILD often limits patient survival. The aim of our study is to evaluate the clinical characteristics and predictors of lung function changes in SSc-ILD patients treated in a real-world setting. (2) Methods: All SSc-ILD cases previously confirmed by rheumatologists and a multidisciplinary ILD team between January 2017 and June 2019 were included (*n* = 54). The detailed medical history, clinical parameters and HRCT were analyzed. The longitudinal follow-up for pulmonary symptoms, functional parameters and treatment were performed for at least 2 years in no treatment, immunosuppression and biological treatment subgroups. (3) Results: In SSc-ILD patients (age 58.7 ± 13.3 years, 87.0% women), the main symptoms included dyspnea, cough, crackles and the Raynaud’s phenomenon. The functional decline was most prominent in untreated patients, and a normal body mass index (BMI < 25 kg/m^2^) was associated with a significant risk of deterioration. The majority of patients improved or were stable during follow-up. The progressive fibrosing-ILD criteria were met by 15 patients, the highest proportion being in the untreated subgroup. (4) Conclusions: SSc-ILD patients who are overweight are at a lower risk of the functional decline and progressive phenotype especially affecting untreated patients. The close monitoring of lung involvement and a regular BMI measurement are advised and early treatment interventions are encouraged.

## 1. Introduction

One of the systemic autoimmune rheumatic disorders—systemic sclerosis (SSc)—is often associated with interstitial lung disease (ILD). In SSc, an interaction of inflammation, vasculopathy, endothelial and fibroblast dysfunction leads to skin thickening, fingertip lesions, Raynaud’s phenomenon and different internal organ manifestations. SSc has been divided according to its 3 phenotypes, which are limited cutaneous SSc (lcSSc, skin sclerosis that develops distal to the elbow and knee), diffuse cutaneous SSc (dcSSc, skin involvement that extends proximally to the elbow and knee and to the body surface with internal organs being affected early) and sine scleroderma SSc (ssSSc, whose internal organ manifestations and serological abnormalities are without skin involvement) [1]. Severe ILD is in general associated with dSSc and anti-SCL-70 antibodies, while it is relatively rare among individuals with anti-centromere antibodies [1,2]. Inflammatory cells accumulate in the lung parenchyma causing alveolar damage, which activates fibroblastic and myofibroblastic cells to produce extracellular matrices causing the scarring of the respiratory system, including diffuse parenchymal lung injury, frequently taking the form of pulmonary fibrosis, pulmonary hypertension and pleural involvement. Gastroesophageal reflux, malignancy, respiratory muscle weakness and heart failure are indirect pulmonary complications in SSc-ILD. Therefore, ILD is a common and usually early comorbidity of SSc, placing a significant burden on both the patient and the health care system, and can be the main cause of mortality [3,4,5,6].

The first symptoms of SSc-ILD are generally exertional dyspnea, fatigue and a dry cough, as an auscultation finding fine crepitation is often present [1]. The diagnosis of SSc-ILD is made by a multidisciplinary ILD-team (MDT), which is represented by respiratory, radiology, pathology specialists and often includes rheumatologists [4]. The presentation of SSc-ILD includes different types of ILDs and can also show different forms of progression. A slow decline in lung function (LF) and CO diffusion parameters can be present with periods of stability or improvement, but rapid progression and severe deterioration can also occur [3,7].

The definition of progressive fibrosing (PF)- ILD describes LF decline associated with either the worsening of respiratory symptoms (mainly dyspnea and cough) and/or a progression of fibrosis on high-resolution computed tomography (HRCT) [8,9,10]. Compared to previous years, nowadays we have a wide range of therapeutic options available. Immunosuppressive (ISU) drugs, such as cyclophosphamide (CYC) and mycophenolate mofetil (MMF), have shown to improve lung function and forced vital capacity (FVC) according to randomized clinical trials [9,11,12,13,14]. In progressive forms of SSc-ILD (PF-ILDs), the focus has shifted towards the tyrosine kinase inhibitor nintedanib, which was approved for progressive SSc-ILD treatment by the European Medicines Agency in 2020 [15,16]. While the co-treatment of MMF and nintedanib is not authorized in this subgroup, the combined therapy showed a more promising outcome, without increase in adverse events in the SENSCIS (A Trial to Compare Nintedanib with Placebo for Patients with Scleroderma Related Lung Fibrosis) trial [15]. High-dose corticosteroids are not recommended in SSc-ILD as first line treatment. In a smaller number of studies, the safety and beneficial adverse reaction profile of rituximab was demonstrated with a similar efficacy as CYC in SSc-ILD [17]. In some centers, bone marrow or lung transplantation is available in extremely carefully selected cases [14].

Measuring the body mass index (BMI) is a useful and simple way to evaluate excess weight. The role of weight loss and BMI in the disease prognosis is already well known in other respiratory diseases, but has not been studied in SSc-ILD [18,19,20,21,22,23,24,25].

Our goal is to assess the clinical course and risk factors for progression in SSc-ILD using longitudinal data on pulmonary evaluation.

## 2. Materials and Methods

### 2.1. Study Population

Our study is based on the retrospective data analysis of SSc-ILD outpatients classified under ICD10 J84 and M34 codes, who were presented at the Semmelweis University Pulmonology Department between January 2017 and June 2019. The diagnosis of SSc and treatment initiation was made by immunology and rheumatology specialists in immunological-rheumatological centers in central Hungary. The majority of patients had dcSSc (93%), while 4 patients had lcSSc, which had been known for more than 10 years in all cases. The dermatological care of patients with skin involvement took place in dermatology centers, and our study did not cover the exact assessment of these symptoms. All patients were evaluated by our MDT. A total of 54 SSc-ILD patients were identified and 42 of them had pulmonary follow-up in the given time interval and were included in the longitudinal evaluation. In accordance with the therapy, the patients were divided into 3 subgroups: no treatment (*n* = 12), immunosuppressive therapy (ISU, *n* = 21) and biological therapy (rituximab or tocilizumab) recipients (*n* = 9). The patient inclusion is summarized in Figure 1. The most frequently applied ISU therapies were CYC or MMF with or without low-dose glucocorticosteroids.

SSc was previously diagnosed, based on the internationally accepted criteria of the American College of Rheumatology/European League Against Rheumatism Collaborative Initiative (EULAR-ACR), by rheumatology specialists working at rheumatology centers in central Hungary [5].

The SSc-ILD patients’ data were collected retrospectively, based on the medical records. In each case, a detailed medical history was attained regarding the pulmonary symptoms, exposures and comorbidities. According to the WHO definition, the patients with a BMI between 25.0–29.9 kg/m^2^ were considered as overweight. All the subjects underwent physical examination, chest X-ray at their pulmonary evaluation. Pulmonary function tests included the measurement of FVC, forced expiratory volume in 1 s (FEV1), FEV1/FVC and total lung capacity (TLC) according to the American Thoracic Society and European Respiratory Society (ARS/ETS) guidelines. For measuring the diffusing capacity for carbon monoxide (DLCO), we used the single breath CO method. In addition, the transfer coefficient of the lung for carbon monoxide (KLCO) was calculated (PDD-301/s, Piston, Budapest, Hungary). Gender–age–physiology (GAP) used in idiopathic pulmonary fibrosis (IPF) to estimate mortality was calculated as a potential risk assessment tool [26]. Patients were tested for physical activity using the 6 min walk test (6MWT). Arterialized capillary blood gases (ABGs) were evaluated at room temperature (Cobas b 221, Roche, Hungary). The Borg scale referring to dyspnea was used.

All patients had been confirmed as ILD by high resolution computed tomography (HRCT), performed in both inspiration and expiration using Philips Ingenuity Core 64 and Philips Brilliance 16 CT scanners. HRCT patterns, such as non-specific interstitial pneumonia (NSIP), usual interstitial pneumonia (UIP) and probable UIP (pUIP) and percentage involvement, were determined by MDT experts according to a visual scoring system [27]. The radiologic features of NSIP include ground-glass opacities (GGO), reticular opacities and traction bronchiectasis in the fibrotic subtype [28]. The UIP pattern comprises subpleural and basal predominance reticulation accompanied by honeycombing and traction bronchiectasis (which can be associated with presence of GGO) probable UIP (pUIP) are the same abnormalities without honeycombing [29,30].

The long-term care is comprised of pulmonological and rheumatological controls in correlation with the patients’ disease requirements. The follow-up (and also treatment time) of at least 31 months (in the longest case 53 months) until June 2021 included radiological controls, therapy management, LF, DLCO and KLCO tests. In this analysis, PF-ILD was defined as a FVC relative annual decline ≥5% in addition to either a deterioration of clinical symptoms or a progression of fibrosis on HRCT, as mentioned previously [10,31].

### 2.2. Statistical Analysis

Statistical analysis was performed using Graph Pad software (GraphPad Prism 5.0 Software, Inc., La Jolla, CA, USA) and Microsoft Excel (Microsoft Corporation, Redmond, DC, USA). Continuous variables were expressed as the mean  ±  standard deviation. The Kolmogorov–Smirnow test was used for the normality test of data. Differences between the groups for continuous data were evaluated in normally distributed data with a Student′s *t*-test. Otherwise the Mann–Whitney U test was used. Analysis of variance (ANOVA) and Tukey’s post hoc analysis were used to examine the continuous variables in the therapeutic subgroups. The chi-squared test and two-tailed Fisher’s exact test were applied for comparing the categorical variables. Predictors of the progression were analyzed using the odds ratio and plot analysis. The correlation between the BMI and FVC was analyzed by logarithmic transformation. All % values are expressed for the whole study population (all patients) or respective subgroups as indicated. A *p*-value < 0.05 was defined as statistically significant.

## 3. Results

The patient characteristics are summarized in Table 1. The mean age of the studied patients was 59 years, the majority of them being women. Non-smokers accounted for 74.1% of the case numbers. Ninety-five percent of patients had 0–3 points (Stage I) at GAP staging, while only 2 patients were in Stage II. The lowest BMI values were in the treatment-free subgroup, the highest in the ISU subgroup. The main symptoms included dyspnea, crackles and Raynaud’s phenomenon, followed by cough, joint pain and digital clubbing. Significantly fewer patients reported cough in the ISU therapy subgroup compared to other subgroups. Gastrointestinal involvement was most often present in the treatment-free group. The most common HRCT radiological pattern is NSIP (predominantly less than 20% of lung involvement), followed by UIP or pUIP (approximately balanced involvement below and above 20%) (Table 1.). In the ISU subgroup, significantly more patients had a NSIP pattern, while, from the patients receiving biological therapy, a pUIP/UIP pattern was predominant. LF data at baseline is summarized in Table 2.

Patients were characterized by mild restrictive functional decline. Annual FVC decline was more distinct in patients who did not receive therapy (−10.2 ± 13.0%), in comparison to patients who underwent ISU (−3.9 ± 5.1%) and biological treatment (−1.04 ± 7.8%). Patients receiving biological treatment showed the slightest degree of FVC deterioration, with 4 patients even showing an improvement. There was no difference in LF between the longitudinal subgroups, however annual FVC decline was less progressive in the treated subgroups. A decrease in DLCO is present in all three observed treatment subgroups, however the relative DLCO decline is greater with patients who received biological therapy (Figure 2). Out of the patients with longitudinal data, 15 patients met our PF-ILD criteria, while 27 did not show signs of deterioration during follow-up. The highest proportion of PF-ILDs was confirmed in the treatment-free subgroup, while over 3/4 of patients were stable or improved during the ISU and biological treatments.

The factors favoring functional stability are being overweight (according to the definition of World Health Organization (BMI > =25 kg/m^2^)) and an absence of anti-SCL-70 positivity (Figure 3). Being overweight was mainly present in the biological treatment subgroup (55.6%) and affected the least patients in the treatment-free subgroup (25%). A significantly lower BMI is present in PF-ILD compared to stable SSc-ILD patients (22.9 kg/m^2^ vs. 25.71 kg/m^2^; *p* = 0.03) ,and a clear negative correlation between baseline BMI and annual FVC reduction (r = −0.97, r^2^ = 0.93, *p* < 0.001) can be established (Figure 4).

A total of 10 patients had gastrointestinal involvement (mostly esophagus dysmotility) with a predominantly low or normal BMI (*p* = 0.019), compared to overweight patients. No significant association was found between gastrointestinal symptoms and PF-ILD.

## 4. Discussion

SSc-ILD has a variable disease course from slow decline with stable periods or improvement to rapidly progressive clinical course. The novelty of our study is that baseline overweight might be an important indicator for more favorable outcomes regarding PF-ILD.

In our study, 15 patients (35.7%) met our PF-ILD criteria, which is in correspondence with international data [1,3,18,19,20]. In 2014, Tiffany and colleagues analyzed 27 reviews on the topic of predictors of progression and mortality in SSc-ILD. In these studies, patient-specific, ILD-specific, and SSc-specific variables predicted the progression and mortality in SSc-ILD [32]. Based on this data, the major risk factors for progressive SSc-ILD include an older age, male sex, extent of lung involvement on initial HRCT, decreased DLCO and FVC at baseline. Our study could not quite verify these factors of PF-ILD, presumably due to the small sample size and the resulting large standard deviation. The presence of anti-Scl-70 antibodies and the absence of anti-centromere antibodies are also widely accepted negative predictors., and this association was also confirmed in our research [32,33].

In accordance with our study, the negative association between functional progression and BMI is a new and clinically important predictive marker in SSc-ILD. The question has to be asked: why is being overweight a protective factor in SSc-ILDs? The negative prognostic role of a lower BMI and weight loss is already known in IPF [18,19,20,21,22]. The post hoc analysis of INPULSIS, CAPACITY and other clinical trials confirm this condition, in which some association was found between the annual FVC decline and BMI at baseline [34,35]. However, contradictory data also exists [36,37], and some key studies, such as SLS (Scleroderma Lung Study) I and II, did also not include the BMI data [12,13]. The aggravating role of a lower BMI and weight loss are described in other respiratory diseases associated with chronic hypoxia, for example, in chronic obstructive pulmonary disease (COPD) [23]. While obesity in COPD appears to be a protective factor (obesity paradox) despite the comorbidities, there is only a limited amount of data about the comorbidities of extreme obesity due to the low number of obese patients in most large COPD studies [24]. The cause might be an electron transport chain dysfunction, which causes decreased muscle stamina in low BMI COPD patients. Considering the concept, a similar phenomenon may occur in SSc-ILD [25]. As it is well known, a lower BMI might be caused by gastrointestinal tract (GIT) involvement [38,39]. However, there was no significant difference in dysphagia and gastroesophageal reflux between the PF-ILD and non-PF-ILD forms, whereas more patients had GIT symptoms with normal BMI.

In our study, we investigated patient characteristics and progression according to therapy. The first line treatment of SSc-ILD is immunosuppression (mainly CYC and MMF) based on the results of previous clinical trials: SLS I and SLS II [11,12,13,40]. More recent biological therapies, such as rituximab and tocilizumab, have expanded the options of treatment [17,41,42]. Due to the risk of provoking renal crisis, the use of glucocorticosteroids in scleroderma is limited to low doses [14]. During the follow-up period, low dose steroid treatment was used only in the ISU subgroup. Although the weight-gaining side effect of glucocorticosteroids is well known and the mean BMI was the highest in the ISU subgroup, the overweight patients were more prevalent in the biological therapy subgroup. Additionally, the PF-ILD ratio was the same in the ISU and biological therapy subgroups (33.3%). The annual decrease in LF was more pronounced in the treatment-free subgroup and, simultaneously, the lowest BMI values were represented in these patients. The close monitoring of these patients (including regular body weight and BMI measurement) is the only way to initiate early treatment in order to intervene and slow down progression. It is important to emphasize that patients with preserved lung function and a normal BMI were less likely to be treated with ISU or biological therapy, and were the most prone to lung function loss. The choice of systemic treatment is decided by rheumatologists and immunologists. The systemic treatment of patients in our research was also carried out by immunologists or rheumatologist in an outpatient setting. However, in patients with mild symptoms, systemic treatment was not considered as necessary. Thus, small disease progression between visits was not noticeable, leading to late therapy initiation. The correct introduction of systemic therapy is, to date, not mentioned in international recommendations [43,44,45]. More attention should be given to patients with good lung function and smaller extent lung involvement on HRCT, especially to patients with a normal BMI, as they are more prone to unnoticed functional decline. The Goh criteria set a low FVC treatment threshold (<70% predicted) that might result in severe functional loss in SSc-ILD patients before treatment initiation. In contrast, monitoring the functional decline and the earlier start of ISU or biological treatment might reduce the risk of deterioration. This concept of early treatment introduction was also pointed out in the EUSTAR database, although BMI was not included as a risk factor of functional decline in their evaluation [7]. Naturally, due to the comorbidities of being overweight, we do not recommend achieving obesity. The target BMI level may be in the upper normal BMI range, but this assumption requires further investigation.

The limitations of our study are the retrospective single center design and the low number of patients. Prospective studies should be encouraged to evaluate the correlation between a lower BMI, weight loss and SSc-ILD. Long term longitudinal follow-up, expert SSc-ILD evaluation by MDT and treatment defined subgroup analysis were the assets of our study.

## 5. Conclusions

BMI is an important prognostic factor in SSc-ILD progression and an initial low and normal BMI or weight loss should be critically followed by the clinical care team. The thorough monitoring of BMI in clinical practice is required and especially patients with normal BMI should be followed closely for deterioration. The timing of the introduction of ISU and biological therapy remain a major challenge for clinicians. A higher awareness and possibly lower threshold for therapy initiation is needed in patients with preserved lung function and a normal BMI, even in clinically or radiologically limited SSc-ILD as functional decline was the most prominent in untreated patients. Although it is important to note that patients with a normal BMI were presented more often with GIT symptoms, GIT involvement was not associated with PF-ILD. Regular LF and BMI measurements should be performed in all SSc patients starting from the initial diagnosis, and anti-SCL-70 positivity is helpful when considering therapy introduction. Future studies are needed to determine the optimal start and combination of therapy.

## Figures and Tables

**Figure 1 biomedicines-10-00434-f001:**
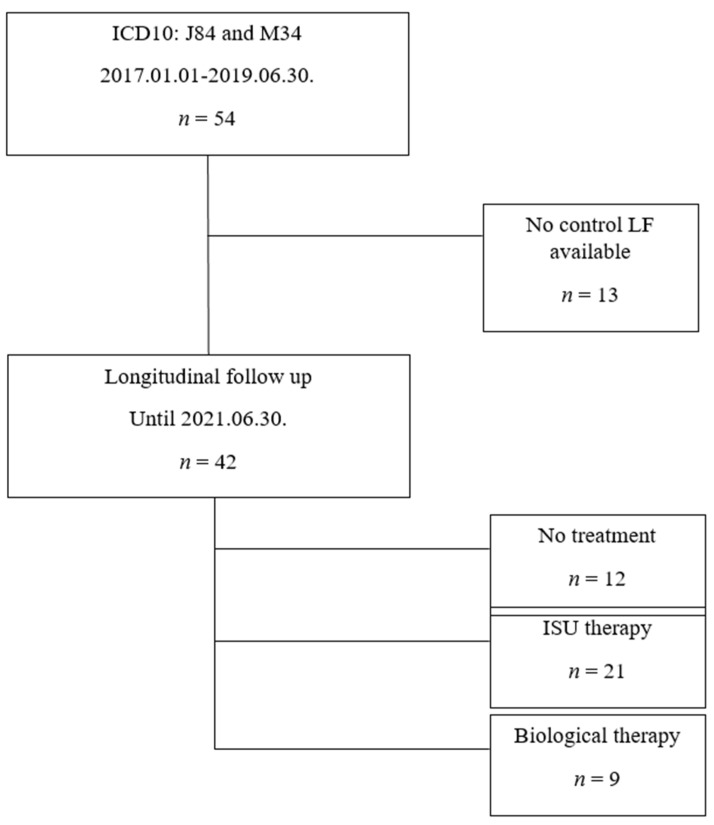
Study population.

**Figure 2 biomedicines-10-00434-f002:**
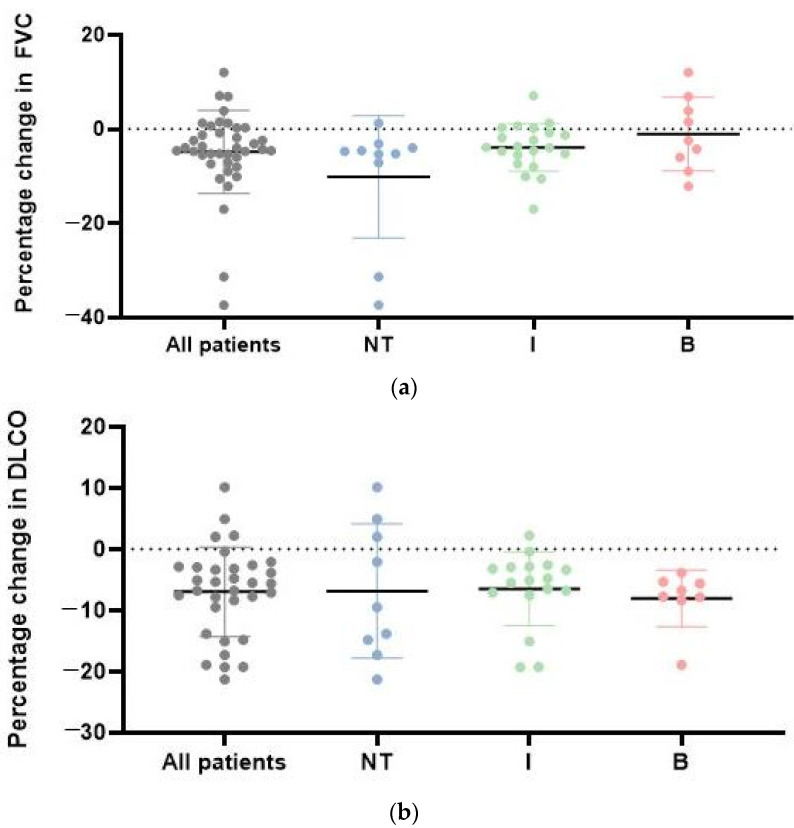
(**a**) Annual FVC changes in all SSc-ILD patients and in the specific treatment groups. Description of what is contained in the first panel; (**b**) annual DLCO changes in each specific treatment group; DLCO, diffusing capacity for carbon monoxide; FVC, forced vital capacity; I = ISU therapy; NT = No treatment; I = ISU therapy, B = Biological therapy.

**Figure 3 biomedicines-10-00434-f003:**
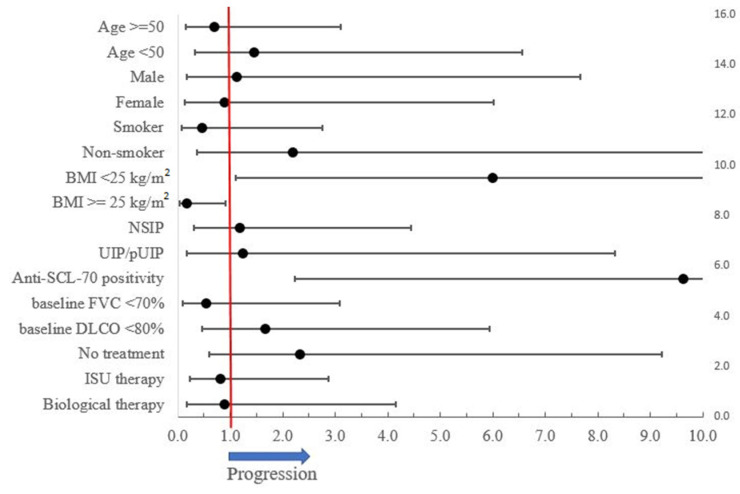
Risk factors of PF-ILD progression.

**Figure 4 biomedicines-10-00434-f004:**
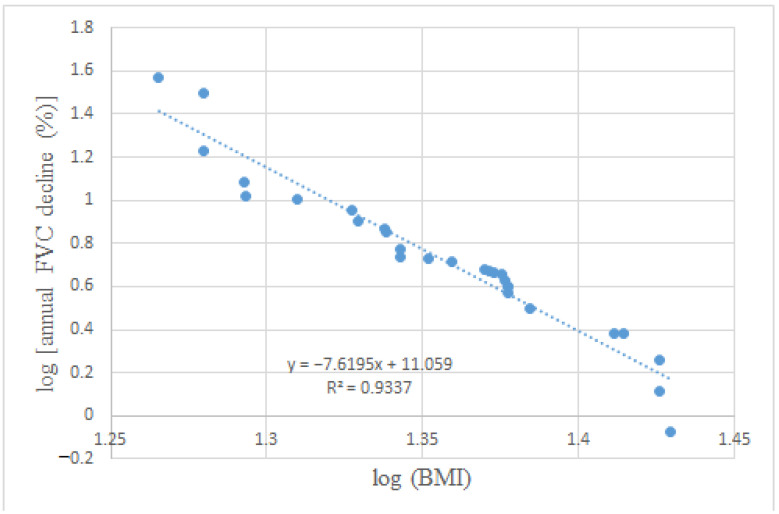
Negative correlation between BMI and annual FVC decline (%)**.** BMI, body mass index; FVC, forced vital capacity.

**Table 1 biomedicines-10-00434-t001:** Patient characteristics.

Parameters	All Patients (*n* = 54) #	No Treatment (*n* = 12)	ISU Therapy (*n* = 21)	Biological Therapy (*n* = 9)
Age (year)	58.7 ± 13.3	66.1 ± 13.7	59.12 ± 12.4	62.7 ± 10.0
Sex (male: female) *n* (%)	7:47 (13.0:87.0)	1:11 (8.3:91.7)	3:18 (14.3:85.7)	1:8 (11.1:88.9)
GAP score *n* (%)				
Stage I (0–3 points)	48 (88.9)	12 (100)	20 (95.2)	8 (88.9)
Stage II (4–5 points)	6 (11.1)	0	1 (5)	1 (1.1)
Stage III (6–8 points)	0	0	0	0
Ever smoker *n* (%) Non-smoker *n* (%)	14 (25.9) 40 (74.1)	4 (33.3) 8 (66.7)	5 (23.8) 16 (76.2)	2 (22.2) 7 (77.8)
BMI (kg/m^2^)	24.8 ± 4.3	23.6 ± 3.1	25.0 ± 4.4	26.4 ± 4.5
Overweight *n* (%)	21 (38.9)	3 (25.0)	8 (38.1)	5 (55.6)
PF-ILD *n* (%)	15 (27.8)	5 (41.7)	7 (33.3)	3 (33.3)
Symptoms *n* (%)				
Dyspnea	26 (48.2)	6 (50.0)	8 (38.1)	5 (55.6)
Cough	15 (27.8)	4 (33.3)	1 (4.8) *	4 (44.4)
Chest pain	5 (9.3)	0	0	0
Joint pain	8 (14.8.)	1 (8.3)	5 (23.8)	1 (11.1)
Clubbing	1 (1.91)	0	0	1 (11.1)
Weight loss	3 (5.6%)	1 (8.3)	1 (4.9)	1 (11.1)
Crackles	15 (27.8)	5 (41.7)	6 (28.6)	2 (22.2)
Raynaud’s phenomenon	38 (70.5)	8 (66.7)	13 (61.9)	5 (55.6)
GIT involvement	10 (18.5)	5 (41.7) ***	2 (9.5)	3 (33.3)
HRCT pattern *n* (%)				
NSIP n	34 (63.0)	11 (91.7)	15 (71.4)	3 (33.3) **
UIP/pUIP n	8 (14.8)	1 (8.3)	2 (9.5)	4 (44.4) ***
Other or no data n	10 (18.5)	0	4 (19.0)	2 (22.2)
Serological pattern *n* (%)				
ANA	23 (42.6)	8 (66.7)	12 (57.1)	3 (33.3)
ACA	1 (1.9)	1 (8.3)	0	0
RF	3 (5.6)	2 (1.7)	1 (4.8)	0
ACCP	1 (1.9)	0	1 (4.8)	0
Anti-RNA-polymerase	2 (3.7)	1 (8.3)	1 (4.8)	0
Anti-cytoplasmatic	5 (9.3)	1 (8.3)	2 (9.5)	2 (22.2)
Anti-chromatin	12 (22.2)	5 (41.7)	6 (28.6)	1 (11.1)
Anti-Smith	1 (1.9)	1 (8.3)	0	0
Anti-Jo-1	1 (1.9)	0	1 (4.8)	0
Anti-SSA	2 (3.7)	0	2 (9.5)	0
Anti-SSB	1 (1.9)	1 (8.3)	0	0
Anti-SCL-70	18 (33.3)	8 (66.7)	9 (42.9) **	1 (11.1)
Anti-RNP	4 (7.4)	3 (25)	1 (4.8)	0
Anti-dsDNA	3 (5.6)	1 (8.3)	1 (4.8)	1 (11.1)

ACA, anticentromere antibodies; ACCP, anti-cyclic citrullinated peptide antibodies; anti-dsDNA, antibodies to double-stranded deoxyribonucleic acid; anti-SSA, Ro autoantibodies; anti-SSB, anti-La antibodies; anti-SCL-70, anti-topoisomerase I antibodies; anti-RNP, antibodies to ribonucleoprotein; ANA, anti-nuclear antibodies; BMI, body mass index; GAP, gender–age–physiology index; GIT, gastrointestinal tract; HRCT, high-resolution computed tomography; ISU, immunosuppressive; NSIP, non-specific interstitial pneumonia; PF-ILD, progressive fibrosing interstitial lung disease; pUIP, probable usual interstitial pneumonia; RF, rheumatoid factor; UIP, usual interstitial pneumonia; # total number of patients were 54, but out of these only 42 had follow-up data; * *p* < 0.05 vs. no treatment and biological therapy subgroup ** *p* < 0.05 vs. no treatment subgroup, *** *p* < 0.05 vs. ISU subgroup.

**Table 2 biomedicines-10-00434-t002:** Lung function, ABG and 6MWT functional parameters.

Parameters	All Patients (*n* = 54)	No Treatment (*n* = 12)	ISU Therapy (*n* = 21)	Biological Therapy (*n* = 9)
Lung function				
FVC (L)	2.5 ± 0.8	2.5 ± 0.8	2.8 ± 0.8	2.2 ± 0.6
FVC (%predicted)	89.8 ± 23.2	97.6 ± 21.7	92.4 ± 26.6	82.2 ± 17.2
FEV1(L)	2.2 ± 0.6	2.1 ± 0.6	2.3 ± 0.7	2.0 ± 0.6
FEV1(%predicted)	90.2 ± 21.8	97.9 ± 21.6	92.2 ± 23.1	85.3 ± 21.5
FEV1/FVC (%)	84.7 ± 6.3	83.4 ± 6.3	84.9 ± 4.9	86.9 ± 12.4
TLC (L)	3.9 ± 1.1	4.0 ± 1.0	4.1 ± 1.4	3.8 ± 1.0
TLC (%predicted)	78.4 ± 21.0	81.4 ± 16.2	81.1 ± 23.0	79.2 ± 23.4
Diffusion parameters				
DLCO (mmol/min/kPa)	5.9 ± 2.0	6.2 ± 1.8	6.2 ± 2.1	5.1 ± 1.3
DLCO (%predicted)	75.2 ± 22.0	86.9 ± 24.5	77.4 ± 20.7	66.7 ± 14.7
KLCO (mmol/min/kPa/L)	1.4 ± 0.4	1.3 ± 0.3	1.4 ± 0.4	1.3 ± 0.3
KLCO (%predicted)	70.0 ± 18.1	66.0 ± 15.6	70.5 ± 18.9	67.0 ± 16.7
ABG				
pH	7.4 ± 0.0	7.4 ± 0.0	7.4 ± 0.0	7.4 ± 0.1
pCO_2_ (mmHg)	38.0 ± 4.7	34.4 ± 3.1	43.4 ± 11.1	36.7 ± 3.6
pO_2_ (mmHg)	74.2 ± 10.5	84.2 ± 13.1	71.2 ± 12.1	72.2 ± 14.6
6MWT				
Distance (m)	444.2 ± 119.8	365.3 ± 233.9	468.8 ± 108.0	342.0 ± 106.6
SpO_2_ baseline (%)	94.9 ± 2.8	96.7 ± 3.2	97.3 ± 2.1	93.9 ± 4.4
SpO_2_ post-exercise (%)	89.9 ± 10.0	82.7 ± 19.9	95.3 ± 2.4	87.6 ± 8.1
Desaturation (%)	4.9 ± 9.3	14.0 ± 16.6	2.5 ± 2.4	8.0 ± 6.3
Pulse baseline (1/min)	84.8 ± 14.6	78.7 ± 11.9	82.9 ± 9.6	86.4 ± 11.6
Pulse post-exercise (1/min)	108.5 ± 23.1	100.3 ± 29.5	108.4 ± 17.6	106.7 ± 26.4
Borg scale baseline (0–10)	0.2 ± 0.5	1.7 ± 2.9	0.1 ± 0.3	1.0 ± 1.4
Borg scale post-exercise (0–10)	1.8 ± 2.5	2.7 ± 3.8	1.6 ± 1.2	3.1 ± 2.5

6MWT, 6 min walk; ABG, arterialized capillary blood gases; DLCO, diffusing capacity for carbon monoxide; FVC, forced vital capacity; FEV1, forced expiratory volume in 1 s; ISU, immunosuppressive; KLCO, transfer coefficient of the lung for carbon monoxide; pCO_2_; partial pressure of carbon dioxide; pO_2_, partial pressure of oxygen ; SpO_2_, oxygen saturation; TLC, total lung capacity.

## Data Availability

The original contributions presented in the study are included in the article; further inquiries can be directed to the corresponding author.

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
