# Peer review of "Treatment and Systemic Sclerosis Interstitial Lung Disease Outcome: The Overweight Paradox"

_biomedicines, 2022, doi:10.3390/biomedicines10020434_

Round 1

Reviewer 1 Report

in this original work the Authors present the result of an observational study on PF changes in a small cohort of SSc patients. In the introduction they should better clarify and support their ideas. They should better introduce the SSc itsel and the caractheristcs of ILD-SSc patients and the significance of lung involvment for the patients prognosis and treatment. In my opinion they should also stratify the patients according to the disease subtype (lSSc or dSSc). it is already known the role of anti-SCl70 antibodies in this subtype. Furthermore they do not mention the skin involvment in this patients. They should also add the mRSS to have a better picture. It s interesting the finding about the protective role of normal weight. To better point out this data they should investigate also the GI involvement in their cohort of patients, in fact the low weight could be related to this disease manifestation, with a more complex patient not only from the pulmonary point of view.

Author Response

Response to Reviewer 1 Comments

Dear Reviewer 1,

Enclosed please find the second revised version of our manuscript entitled Treatment and systemic sclerosis interstitial lung disease out-come: the overweight paradox to publish in Biomedicines.

We are grateful for the effort of the Reviewers and the Editorial Office and we hope that our responses will meet the satisfaction of the journal.

As requested, please find our point-by-point responses to the comments below.

Yours sincerely,

Authors

Introduction, Results and Discussion sections have been revised for clearly understanding.

 Point 1: In this original work the Authors present the result of an observational study on PF changes in a small cohort of SSc patients. In the introduction they should better clarify and support their ideas. They should better introduce the SSc itself and the caractheristcs of ILD-SSc patients and the significance of lung involvment for the patients prognosis and treatment.

Response 1: The Reviewer has a good point. We have rephrased the sentences in the Introduction section focusing on the characteristics of SSc and its subtypes,its disease prognosis, its lung involvement, and therole of anti-SCL-70 antibodies, and summarizing treatment options.

Point 2: In my opinion they should also stratify the patients according to the disease subtype (lSSc or dSSc). It is already known the role of anti-SCl70 antibodies in this subtype. Furthermore they do not mention the skin involvment in this patients. They should also add the mRSS to have a better picture.

Response 2: The comment of the Reviewer is valid. Our clinic is responsible for the outpatient care of SSc patients. Immunological and dermatological care took place in other special centers of Hungary. Thus, modified Rodnan skin score results were also not available to us. Ssc phenotypes were described in the Introduction section. For better understanding we have added a paragraph in the Methods section: „The diagnosis of SSc and treatment initiation had been made by immunology specialists in immunological centers in Central Hungary. The majority of patients had dSSc (93%), while 4 patients had lSSc, which had been known for more than 10 years in all cases. The dermatological care of patients with skin involvement took place in dermatology centers, and our study did not cover the exact assessment of these symptoms.”

Point 3: It's interesting the finding about the protective role of normal weight. To better point out this data they should investigate also the GI involvement in their cohort of patients, in fact the low weight could be related to this disease manifestation, with a more complex patient not only from the pulmonary point of view.

Response 3: We fully agree with the Reviewer. We have added GIT involvement into Table 1. We have examined gastrointestinal involvement and we have replaced this fact in the Results, Discussion and Conclusions sections.

Results: “A total of 10 patients had gastrointestinal involvement (mostly esophagus dysmotility) with predominantly low or normal BMI (p = 0.019) compared to overweight patients. No significant association was found between gastrointestinal symptoms and PF-ILD.”.

Dicussion: “As it is well known lower BMI might be caused by gastrointestinal tract (GIT) involvement [37, 38]. However, there was no significant difference in dysphagia and gastroesophageal reflux between the PF-ILD and non-PF-ILD forms, whereas more patients had GIT symptoms with normal BMI.”

Conclusions: “It is important to note that patients with normal BMI presented more often with GIT symptoms, however GIT involvement was not associated with PF-ILD.”

Reviewer 2 Report

In the introduction, it is necessary to specify the interest in the BMI study, because it appears out of nowhere in methodology and then it is its main result.

Table 1 has errors, for example in all patients they have 54, when between the 3 groups they are 42, in ever smokers and non smokers instead of % they have frequencies. In chest pain in total they have a frequency of 5 and in the three groups they have 0, the numbers in dyspnea and cough do not match either, and in the column of all there is a 2 after the parenthesis

Write errors in row 149

The authors do not specify why one group did not receive treatment if the diagnosis was confirmed.

It would be important to know the evolution time in the 3 groups and the treatment time

What is your explanation for this result "Novelty of our study is that baseline normal BMI might be an important negative indicator in PF-ILD", add in the discussion if it will be appropriate to use the BMI or what other study should be used according to what who comment on paradoxical obesity in these patients with SSc-ILD.
Discuss how the use of glucocorticosteroids can influence this increase in BMI. in patients with SSc-ILD.

Would the authors propose that patients have to be at least overweight to prevent progression?

Author Response

Response to Reviewer 2 Comments

Dear Reviewer 2,

Enclosed please find the second revised version of our manuscript entitled Treatment and systemic sclerosis interstitial lung disease out-come: the overweight paradox to publish in Biomedicines.

We are grateful for the effort of the Reviewers and the Editorial Office and we hope that our responses will meet the satisfaction of the journal.

As requested, please find our point-by-point responses to the comments below.

Yours sincerely,

Authors

Introduction, Results and Discussion sections have been revised for clearly understanding.

 Point 1: In the introduction, it is necessary to specify the interest in the BMI study, because it appears out of nowhere in methodology and then it is its main result.

Response 1: We fully agree with the Reviewer and we have added a paragraph about BMI to the Introduction section: Measuring body mass index (BMI) is a useful and simple way to evaluate the excess weight. The role of weight loss and BMI in disease prognosis is already well known in other respiratory diseases, but has not been studied in SSc-ILD [18-25].

Point 2: Table 1 has errors, for example in all patients they have 54, when between the 3 groups they are 42, in ever smokers and non smokers instead of % they have frequencies. In chest pain in total they have a frequency of 5 and in the three groups they have 0, the numbers in dyspnea and cough do not match either, and in the column of all there is a 2 after the parenthesis.

Response 2: Total number of patients were 54, but out of these only 42 had follow-up data, so this is the reason why the numbers in the rows don’t add up. The following character is shown below the table: # Total number of patients were 54, but out of these only 42 had follow-up data. Smokers and non smokers rate was corrected according to parameters: n (%).

Point 3: Write errors in row 149.

Response 3: Spell check has been done.

Point 4: The authors do not specify why one group did not receive treatment if the diagnosis was confirmed.

Response 4: According to the Reviewer’s suggestions Discussion section was revised to specify this question: “Choice of systemic treatment is decided by rheumatologists and immunologists. The systemic treatment of patients in our research was also carried out by immunologists in an outpatient setting. However in patients with mild symptoms systemic treatment was not considered necessary. Thus small disease progression between visits was not noticeable, leading to late therapy initiation. The correct introduction of systemic therapy is as of yet not mentioned in international recommendations[42-44].”

 Point 5: It would be important to know the evolution time in the 3 groups and the treatment time.

Response 5: We have rephrased this paragraph in the Methods section according to the Reviewers suggestion: “Follow-up (and also treatment time) of at least 31 months (in the longest case 53 months) until June 2021 included radiological controls, therapy management, LF, DLCO and KLCO tests.”

Point 6: What is your explanation for this result "Novelty of our study is that baseline normal BMI might be an important negative indicator in PF-ILD", add in the discussion if it will be appropriate to use the BMI or what other study should be used according to what who comment on paradoxical obesity in these patients with SSc-ILD-

Response 6: We have rephrased the sentence above in the Discussion section in the following manner: “The novelty of our study is that baseline overweight might be an important indicator for favourable outcome regarding PF-ILD.” The comment of the Reviewer about using BMI in clinical practice is legitimate, we have completed the Conclusions in the following manner: “Thorough monitoring of BMI in clinical practice is required and especially patients with normal BMI should be followed closely for deteoriation.”

Point 7: Discuss how the use of glucocorticosteroids can influence this increase in BMI. in patients with SSc-ILD.

Response 7: We have added a paragraph in the Discussion section as follows: “Due to the risk of provoking renal crisis, the use of glucocorticosteroids in scleroderma is limited.[14] During the follow-up period low dose steroid treatment was used only in the ISU subgroup. Although the weight-gaining side effect of glucocorticosteroids is well known and the mean BMI was the highest in the ISU subgroup, overweight patients were more prevalent in the biological therapy subgroup. Additionally the PF-ILD ratio was the same in the ISU and biological therapy subgroups (33.3%).”

Point 8: Would the authors propose that patients have to be at least overweight to prevent progression?

Response 8: Good question from thr Reviewer. The answer has been added in the Discussion section: “Naturally, due to the comorbidities of overweight we do not recommend achieving obesity. The target BMI level may be in the upper normal BMI range but this assumption requires further investigation.”

Round 2

Reviewer 2 Report

I consider that the modifications made are adequate.

Author Response

Dear Reviewer 2,

Enclosed please find the revised version of our manuscript entitled Treatment and systemic sclerosis interstitial lung disease out-come: the overweight paradox to publish in Biomedicines.

We are grateful for the effort of the Reviewers and the Editorial Office and we hope that our responses will meet the satisfaction of the journal.

As requested, please find our point-by-point responses to the comments below.

Yours sincerely,

Authors

Point 1: Please correct the claim on lines 43-47: Thus, based on different literature sources, one could discuss, which one is the most common, nevertheless, based on meta-analysis data, SSc is not the most common systemic autoimmune rheumatic disorder. The meta-analyses based prevalence of systemic autoimmune rheumatic disorders is as follows:

- Sjogren syndrome - 61/100 000

- Systemic lupus erythematosus -  36/100 000

- Idiopathic inflammatory myopathies - 25/100 000

- Systemic sclerosis - 17/100 000

 Response 1: We fully agree with the Reviewer and we have rephrased the sentence in the Introduction:

Point 2: Please use the officially established/used abbreviations (throughout the manuscript):

-lcSSc (limited cutaneous SSc)

-dcSSc (diffuse cutaneous SSc)

-ssSSc (sine scleroderma SSc)

Response 2: In agreement with the Reviewer’s suggestions abbreviations have been modified according to the official nomenclature.

Point 3: The assessment of ILD percentage involvement on HRCT should be backed up by a specific reference, in which this method was described (since there are several methods available).

Response 3: We expanded the Methods section (“HRCT pattern like non-specific interstitial pneumonia (NSIP), usual interstitial pneumonia (UIP) and probable UIP (pUIP) and percentage involvement were determined by MDT experts according to a visual scoring system [27].”) and added the appropriate reference (Wangkaew, S., et al., Correlation of delta high-resolution computed tomography (HRCT) score with delta clinical variables in early systemic sclerosis (SSc) patients. Quant Imaging Med Surg, 2016. 6(4): p. 381-390.).

Point 4: Line 144-155: please use a separate sub-heading for Statistical analysis
Response 4: According to the Reviewer’s suggestions subheading for Statistical analysis have been added.

Point 5: In Table 1 and 2: a lot of abbreviations are not included (the list should include all of the mentioned abbreviations in each Table); all abbreviations should be listed in alphabetical order for better orientation; a statement should be included, that data in bold are significant (However, bold highlights are not necessary, since an asterisk highlights the significance as well)

Response 5: We fully agree with the Reviewer. Abreviations have been included in Table 1 and 2. Bold highlights were canceled.

Point 6: Line 181-182: no spacing between the numbers and %.

Response 6: The space has been added.

Point 7: Figure 2: the clarity of the graphs could be improved by modifying the y axis legend as follows:

  1. a) Percentage change in FVC
  2. b) Percentage change in DLCO

Response 7: Figure 2/a and 2/b were improved by modifying the title of y axis as shown above.

Point 8: Line 230: SLS I and II should be first written in full, and the abbreviation then used in the brackets).

Response 8: The Reviewer has a good point, therefore Scleroderma Lung Study was written in full in Line 246 when it was first mentioned.

Point 9: At the end of the manuscript, a list of all abbreviations (ideally listed in alphabetical order) should be included.

Response 9: A list of abbreviations have been included in alphabetical order.

Point 10: Even though the quality of the English language/grammar/syntax has been improved, professional language editing would further improve the quality of this manuscript.

Response 10: As suggested, the English language was revised by a native speaker.
